# The Effect of Resistance Exercise Intensity on Acute Hyperglycemia in Young Adult Males

**DOI:** 10.3390/sports8090121

**Published:** 2020-09-03

**Authors:** Evan E. Schick, Luis E. Segura, Shayán Emamjomeh, Joshua A. Cotter

**Affiliations:** Physiology of Exercise and Sport Laboratory, California State University, Long Beach, CA 90840, USA; lesegura12@icloud.com (L.E.S.); shayan.emamjomeh@gmail.com (S.E.); Joshua.Cotter@csulb.edu (J.A.C.)

**Keywords:** glycemic control, strength training, glucose, insulin

## Abstract

The purpose of this study was to examine the effect of resistance exercise (RE) intensity on acute hyperglycemia, in young adult males. Thirteen males (age 23.43 ± 2.18 years, height 175.16 ± 10.44 cm, weight 77.02 ± 8.91 kg, body fat 20 ± 0.03%) participated in three randomized testing sessions, each separated by 96 h. The three testing conditions were: control (CON; induction of hyperglycemia with no-exercise), high (HI; induction of hyperglycemia followed by high-intensity RE: 5 × 4, 90% 1-repetition maximum (1-RM)), and moderate (MO; induction of hyperglycemia followed by low-intensity RE: 3 × 14, 65% 1-RM). In all three testing conditions, participants orally ingested a D100 (100 g/10 oz) glucose beverage at a dosage of 2 g glucose/kg body weight and capillary blood was obtained for plasma glucose and insulin analysis at 0 (fasting), 30, 60, 90 and 120 min, following glucose ingestion. At 30-min post-ingestion in the HI and MO conditions, participants began the respective RE protocols. Acute hyperglycemia was achieved throughout all three 2-h testing conditions; mean 2-h plasma glucose levels during CON (7.1 ± 1.3 mmol∙L^−1^), MO (7.5 ± 0.6 mmol∙L^−1^) and HI (8.2 ± 1.9 mmol∙L^−1^) were all significantly (*p* < 0.05) greater than mean fasting plasma glucose (5.6 ± 0.46 mmol∙L^−1^). Plasma glucose and insulin did not differ (*p* < 0.05) between treatment conditions at any times points over the 120 min, however, 2-h glucose area under the curve in the HI condition was significantly greater (*p* < 0.05) than CON and MO. In conclusion, this study indicates that hyperglycemia, induced prior to RE, may be exacerbated by high-intensity RE.

## 1. Introduction

Type 2 diabetes mellitus (T2DM) continues to be a global pandemic, with an estimated 500 million cases worldwide [1,2,3]. Although T2DM mostly affects individuals ≥ 45 years-old, a small but growing body of evidence demonstrates an increased prevalence of symptomology related to pre-diabetes, metabolic syndrome and T2DM among younger populations [4,5,6,7,8,9,10,11,12,13]. Together, this body of literature reveals that children, adolescents and even collegiate athletes can exhibit symptoms consistent with impaired glycemic control such as elevated fasting plasma glucose (FPG), hemoglobin A1c (HbA1c) and abdominal obesity [7,9,11,12]. Therefore, young individuals may experience bouts of hyperglycemia on a level previously thought to only affect older, sedentary individuals with pre-diabetes, metabolic syndrome or T2DM. Yet, despite these reports—and the likelihood that the etiology of glycemic dysfunction might start at a much younger age than previously thought—little is understood about the nature of hyperglycemia in younger populations.

Skeletal muscle contributes considerably to proper glycemic control, as it accounts for approximately one-third of dietary glucose disposal, and contraction can increase skeletal muscle glucose transport several-fold [14,15,16,17,18,19]. While resistance exercise (RE) is the most effective means of maintaining mass and strength of skeletal muscle [20,21], previous studies have demonstrated that RE, particularly when done at higher intensity, can cause acute bouts of hyperglycemia, even in healthy euglycemic individuals [22,23,24]. This appears to be primarily mediated by stress-induced amplification of the sympathoadrenal drive and resultant increases in hepatic glucose production [25,26]. Little is understood about the health implications of acute hyperglycemia in healthy populations, yet, prior data showing a clear presence of signs related to impaired glycemic control in young and athletic populations raises concern that acute bouts of RT-induced hyperglycemia may not be completely benign for long-term health.

High glycemic index (HGI) foods and beverages are commonly consumed by individuals shortly prior to engaging in RE, as they are believed to improve strength, power and endurance throughout bouts of RE [27,28,29,30,31]. Since HGI fueling makes glucose more readily available for exercise, consumption of HGI food and beverages immediately prior to RE may cause individuals to commence bouts of RE while already hyperglycemic [32,33]. Provided the aforementioned evidence that: (1) younger populations are not immune to the symptomology of impaired glycemic control, (2) RE can cause hyperglycemia in otherwise healthy, euglycemic individuals and (3) HGI food consumption prior to RE is common and may lead to acute bouts of hyperglycemia immediately preceding the onset of RE (pre-RE hyperglycemia), it is critical to understand how RE affects blood glucose in young, heathy individuals who are hyperglycemic at the start of RE. If certain RE parameters do indeed exacerbate pre-existing hyperglycemia in a young and healthy population, it is then paramount for any active individual to be aware of his/her health and nutritional status prior to engaging in intense activity, as any undiagnosed symptoms of glycemic impairment could be worsened.

In an effort to address this gap in knowledge, our group recently published a novel method for inducing pre-RE hyperglycemia in healthy, young individuals [34]. In short, we demonstrated that hyperglycemia, induced 30 min prior to RE, was maintained for two hours (mean two-hour plasma glucose = 6.7 mmol·L^−1^) during tests of maximal muscular strength and power. This model successfully mimics the estimated average glucose (eAG) in prediabetes (HbA1C of ≥ 5.7%; eAG ≥ 6.5 mmol·L^−1^) [35]. Furthermore, this acute bout of hyperglycemia did not significantly impact basic expressions of muscular function like maximal strength, power and endurance. It is imperative however, to examine how alterations in exercise program variables that govern the adaptive muscular response to RE may affect blood glucose in young, healthy individuals who were hyperglycemic at the start of RE. Since RE intensity is a commonly manipulated acute program variable in most training programs, the present study sought to compare the effect of RE intensity on pre-RE hyperglycemia in healthy, young adult males. 

## 2. Materials and Methods

### 2.1. Participants

Thirteen healthy males (age 23.43 ± 2.18 years, height 175.16 ± 10.44 cm, weight 77.02 ± 8.91 kg, body fat 20 ± 0.03%) voluntarily participated in the study (Table 1). Participants were included if they had ≥ six months resistance-training experience. Additionally, participants had to be free of all diagnosed orthopedic, cardiovascular, respiratory and metabolic conditions and were excluded if they incurred an unresolved orthopedic injury within 12 months of the study. Participants over the age of 35 were excluded from the study as glucose metabolism deteriorates with age. All subjects signed a written informed consent and filled out a detailed health history questionnaire prior to testing. Participants were recruited from California State University, Long Beach (CSULB). All participants filled out a Physical Activity Readiness Questionnaire (PAR-Q) and an informed consent form before the study commencement. All testing procedures were approved by the CSULB Institutional Review Board (ID: 1133900-3).

### 2.2. Protocol

All participants underwent four total sessions separated by 96 h, including an initial familiarization session, during which 1-RM testing for both bench press (BP) and back-loaded half squat (BHSQ) was conducted. Familiarization was followed by three randomized experimental treatment sessions: Control (CON), High Intensity (HI) and Moderate Intensity (MO). Please see Table 2 below for a visual organization of the conditions. Familiarization sessions took between 60–90 min at the Physiology of Exercise and Sport (PEXS) Laboratory. The following measurements were collected during the participant’s familiarization session: body weight (DigiTOL, Mettler Toledo), height (SECA, Hamburg, Germany), body composition (Dual Energy X-ray Absorptiometry; Lunar DPX-IQ Lunar Radiation Corp., Madison, WI, USA) and muscular strength. For muscular strength, each participant performed a 1-RM strength test for the BP, followed by the BHSQ on Rogue Monster Rack Series-4 (Rogue Fitness, Columbus, OH). The 1-RM BP and BHSQ protocols were constructed and followed in accordance with the standards set by the National and Strength and Conditioning Association [36]. Briefly, participants first performed a general warm-up consisting of 5 min on a cycle ergometer at a self-selected intensity. Next, participants completed five to ten repetitions of either BP or BHSQ at a light resistance (<30% projected 1-RM) as part of a specific warm-up. Following a one-minute rest period, three to five more repetitions were performed with an additional 5–10 lbs for BP and 30–40 lbs for BHSQ. Participants then rested for two minutes before performing two to three repetitions with an additional 5–10 lbs or 30–40 lbs respectively. All subsequent sets included two-min rest periods and only single repetitions with incremental increases between 10–40 lbs until the 1-RM was completed. Load was incrementally added to achieve the 1-RM within 3–5 sets. 

In the CON, HI and MO conditions (Table 2), participants arrived at the laboratory between 0700–0900 h following an overnight fast. In all three conditions, blood was first collected for fasting plasma glucose and insulin analysis, and then glucose was orally ingested at a dosage of 2 g/kg bodyweight (BW) (Fisherbrand 100 g D-Glucose). Our group previously demonstrated that an oral glucose dosage of 2 g/kg BW achieves a mean plasma glucose of 6.7 mmol∙L^−1^ for two hours [34], which mimics eAG in prediabetes (eAG ≥ 6.5 mmol∙L^−1^) [37]. In the CON condition, participants were then instructed to stay seated for the remainder of the 2 h while capillary blood was collected in 30-min increments for glucose and insulin analysis. In the HI and MO conditions, participants began the respective RE protocols 30 min after glucose ingestion. 

BP and BHSQ were chosen for the RE conditions as an efficient means of recruiting extensive amounts of muscle mass in the upper and lower body, respectively. HI, 90% 1-RM load was used to perform five sets of four repetitions of BP followed by BHSQ. Rest periods were set at three minutes between sets and four minutes between the two exercises. For MO, 65% 1-RM load was used to perform three sets of 14 repetitions of BP and BHSQ. Rest periods were set at two-minutes between sets and three minutes between the two exercises. Total volume loads per intervention were calculated by multiplying the weight lifted per participant (kg) by the number of repetitions and sets completed. For BP, subjects were situated on the flat bench so that they maintained five points of contact (head, upper back, gluteus and feet). Grip width was required to be at least slightly beyond shoulder width, while the experimenter instructed an eccentric cadence of 2 s and a concentric cadence of one second [38]. For BHSQ, subjects were allowed to choose bar placement (low or high bar) although foot position was required to be at least shoulder width. Cadence was instructed at three seconds for eccentric and one second concentric [39]. Depth of the BHSQ, which was monitored by visual inspection, was sufficient when the femur reached parallel to the floor (half squat).

### 2.3. Biochemistry

Finger-prick blood collection took place at 0 (fasting), 30, 60, 90, and 120 min post-glucose ingestion using disposable BD Genie™ spring-loaded lancets (Becton Dickinson & Co, Franklin Lakes, NJ, USA). Plasma glucose was analyzed via StatStrip Xpress^®^ 2 glucometer (Nova Biomedical, Waltham, MA, USA). Immediately following glucose testing, approximately 250–300 μL of whole capillary blood was collected from the same finger prick wound into EDTA-coated capillary tubes (Microvette^®^ CB 300 K2E, Sarstedt Aktiengeselischaft & Co., Sarstedt, Germany) and immediately centrifuged at 14,000 revolutions per minute at 4 °C for 10 min (Heraeus™ Fresco™ 17 Thermo Fisher Scientific Inc., Waltham, MA, USA). Approximately 100 μL of plasma was extracted and stored at −20 °C for insulin analysis. Insulin was analyzed according to manufacturer procedures using a Quantikine^®^ ELISA kit (R & D Systems, MN, USA; Miller, Ruby, Laskin, & Gaskill, 2007). 

### 2.4. Statistics 

All data analysis was performed with IBM^®^SPSS Statistics 25.0 (IBM, Chicago, IL, USA) with significance set at *p* < 0.05. A two-way 3 × 5 (treatment (CON, MO, HI) × time (0, 30, 60, 90, 120 min)) repeated measures analysis of variance (ANOVA) was run to determine the effect of different treatments over time on circulating plasma glucose and insulin concentrations. A one-way ANOVA was implemented to determine if any statistical significance was found in area under the curve (AUC) for total glucose and insulin. Pearson product-moment correlation was run to assess the relationship between Lean Body Mass (kg) and AUC for glucose and insulin. 

## 3. Results

### 3.1. Induction of Pre-RE Hyperglycemia

The protocol successfully induced acute hyperglycemia throughout each of the three two-hour testing sessions, as mean two-hour plasma glucose levels for CON (7.1 ± 1.3 mmol∙L^−1^), MO (7.5 ± 0.6 mmol∙L^−1^) and HI (8.2 ± 1.9 mmol∙L^−1^) were all significantly (*p* < 0.05) greater than mean fasting plasma glucose (5.6 ± 0.42 mmol∙L^−1^) (Table 3). Furthermore, mean two-hour plasma glucose was also significantly greater (*p* < 0.05) for HI than CON. All three of these two-hour mean plasma glucose values were greater than that of our previous study (6.7 mmol∙L^−1^) [34] and HI values (8.2 ± 1.9 mmol∙L^−1^) were in the eAG range for T2DM (≥7.8 mmol∙L^−1^) [37]. 

### 3.2. Glycemic Response during RE

To compare the effect of RE intensity on hyperglycemia, participants were subjected to three testing sessions including, CON (hyperglycemia without RE), MO (hyperglycemia plus moderate-intensity RE) and HI (hyperglycemia plus high-intensity RE). Table 4 summarizes the volume load reached for each lift in the MO and HI conditions. Volume loads were not statistically different (*p* > 0.05) among conditions.

Throughout all three testing sessions, capillary blood was collected in 30-min increments to measure plasma glucose and insulin. During the two-hour testing period, 0 to 30 min was spent passively resting prior to RE in both MO and HI. For MO, exercise occurred between 30–60 min and recovery was 60–120 min. For HI, exercise occurred between 30–75 min and 75–120 min was recovery (Figure 1 and Figure 2). Although the main effects were found for both conditions and time (*p* = 0.015 and *p* = < 0.000 respectively), plasma glucose did not differ significantly between conditions (*p* = 0.146) at any of the time points (Figure 1). However, the total blood glucose response, as calculated by two-h area under the curve (AUC), was significantly greater in HI compared to CON (*p* = 0.012) (Figure 1B). There were no main effects or interactions for plasma insulin and the total insulin response (AUC) was similar between conditions (*p* > 0.05) (Figure 2).

### 3.3. Body Composition and Glycemic Response

A Pearson product-moment correlation revealed a moderately strong negative correlation [40] between lean body mass and two-hour glucose AUC during the MO condition (*r* = −0.78, *p* = 0.002) (Figure 3). 

## 4. Discussion

Although previous studies have examined the impact of RE on blood glucose [22,41,42], the present study is, to the best of our knowledge, the first to induce hyperglycemia prior to RE in order to examine how RE intensity influences pre-RE hyperglycemia. In line with our previous work, oral ingestion at 2 g/kg BW of D-100 glucose solution induced and maintained acute hyperglycemia for two hours in all three conditions (CON, MO and HI) at a magnitude on par with eAG in prediabetes and, in the case of HI, T2DM. Our main finding is that induced hyperglycemia was upheld more effectively by high-intensity RE than by either low-intensity RE or no RE. Secondarily, lean body mass was inversely associated with the total glucose response in the MO condition. 

Recent evidence of abdominal adiposity and fasting hyperglycemia in high school and collegiate athletes suggests that the impact of diabetes may start early in life and that even young, exercise-trained individuals may experience periods of acute hyperglycemia at various times of the day [11,12,13,42]. Furthermore, it is reasonable to believe that athletes of any level may experience acute hyperglycemia prior to resistance training sessions, as the period of time shortly preceding training is a common time for HGI nutritional fueling, including energy drinks and other sugar-containing beverages [29,33]. In our first effort to better understand the nature of acute hyperglycemia and RE, we found that hyperglycemia, induced 30 min prior to RE, had no effect on maximal muscular strength, power or endurance [34]. The present study extended this effort by addressing the impact of RE intensity on acute hyperglycemia, induced pre-RE. Although there were no time point-specific differences in plasma glucose throughout the two-hour MO, HI and CON sessions, the total glucose response, as assessed by two-hour glucose AUC and two-hour mean glucose, was greatest in the HI condition. Similarly, previous reports have shown that blood glucose rises proportionally to the intensity of resistance and endurance exercise in both healthy and type 2 diabetic individuals [22,41,42,43]. The crucial difference between the present study and those aforementioned, is that we induced hyperglycemia prior to exercise while these other studies did not. 

Mechanistically, our data indicate, as others have [22,27,44,45], that high-intensity RE elicits a greater physiologic stress response than low or moderate intensity. Indeed, previous work has illustrated that higher intensity exercise intensifies sympathoadrenal drive, resulting in increased catecholamine and glucocorticoid release [24,26]. Catecholamines and glucocorticoids both enhance hepatic glucose production (HGP)—via gluconeogenesis and glycogenolysis—in order to ensure adequate energy availability for working tissue during the stress period [19,26]. However, during exercise, the impact of HGP on blood glucose excursion is partly counterbalanced by contraction-induced skeletal muscle glucose transport [16,17]. Furthermore, the magnitude of contraction-induced glucose transport is proportional to contraction intensity [17]. Yet, as RE intensity increases, HGP appears to outpace the rate of muscle glucose transport, which induces more exaggerated levels of hyperglycemia during and immediately following RE [24]. Our data show that this intra-RE excursion in blood glucose is exacerbated if an individual comes into the exercise bout already hyperglycemic. However, the present study does not suggest that high-intensity RE should be avoided, since it has been shown to improve post-exercise glucose tolerance and insulin sensitivity in both healthy and T2DM individuals [20,21,44,45,46]. Rather, our data show that the potential for RE to improve glycemic control may be attenuated without proper pre-exercise nutrition practices. What is more, when combined with recent studies showing the existence of symptoms consistent with impaired glycemic control in young and athletic populations [4,5,6,7,8,9], our findings indicate that active individuals, regardless of age, should be aware of their health status prior to engaging in intense activity, as any undiagnosed symptoms of glycemic impairment could be aggravated. 

In demonstrating that high-intensity RE more aggressively perpetuates pre-existing hyperglycemia than low-intensity RE, the present study underscores the importance of utilizing lower glycemic index (LGI) foods prior to engaging in intense bouts of RE, particularly if avoiding extended periods of hyperglycemia is a primary concern. This is especially important for active individuals with prediabetes or T2DM, as they are generally more susceptible to prolonged periods of hyperglycemia [47,48]. Indeed, LGI foods consumed prior to prolonged exercise have been shown to attenuate post-exercise hyperglycemia and hyperinsulinemia and may even enhance endurance-based performance [49,50]. Moreover, in adults with prediabetes, HGI food consumption prior to exercise has been shown to dampen exercise-induced improvements in glucose effectiveness, a measure of the ability of increased plasma glucose to suppress HPG [25]. Importantly, both endurance and resistance-based exercise, performed at high or low-intensity, can reduce periods of hyperglycemia and improve glucose tolerance [27,43,44,45,46,47,48,51]. Yet, our data reveal that these benefits may be mitigated by pre-exercise hyperglycemia. This is a novel finding that necessitates much follow up to better understand how HGI versus LGI feedings prior to exercise impact post-exercise glycemic control in young, healthy individuals. 

In the current study, individuals with a greater lean mass tended to exhibit a lower total glucose response in the LO condition, a finding that exemplifies the powerful glucoregulatory nature of skeletal muscle. Resting skeletal muscle is responsible for approximately one-third of dietary glucose disposal, and contraction can increase skeletal muscle glucose transport several-fold [14,15,16,17,18,19]. Additionally, these data are corroborated by our previous work [34] in which body fat percentage was positively correlated with total glucose response during RE; lean mass was not directly measured in that study. Taken together, these findings suggest that skeletal muscle mass plays a crucial role in maintaining systemic glucose tolerance, even in young and healthy individuals.

This study contains several limitations that should be considered in future research. Firstly, not every subject was able to complete all 14 repetitions in the third set of BP and BHSQ at the moderate intensity level, leading to some variation among subjects in total volume load. We also only used two resistance exercises, and although they both engage large muscle mass, this does not exactly relate to real-world lifting practices. The dosage of glucose used (2 g/kg BW) is greater than the sugar content of pre-workout beverages, thus the glycemic response in this study would be expected to be exaggerated compared to a typical pre-workout response. Finally, post-exercise glucose tolerance and insulin sensitivity were not measured, thus the present study only shows a pre-, intra- and immediately post-workout glycemic response. Therefore, future research should (1) limit the number of repetitions per set at moderate intensity levels in order to better control for inter-subject volume load variability, (2) add more resistance exercises, (3) compare the high glucose dose to that of a typical pre-workout beverage and (4) measure glucose tolerance at 1 and 24 h after exercise.

In summary, our findings illustrate that high-intensity RE perpetuates pre-RE hyperglycemia in young, healthy individuals. These data—in conjunction with prior studies showing that young and athletic populations may exhibit symptoms consistent with impaired glycemic control—demonstrate that, regardless of age, active individuals should be aware of their health status prior to engaging in intense activity, as any undiagnosed symptoms of glycemic impairment could be negatively impacted. We also contribute data suggesting that muscle mass, regardless of physical activity level or age, remains a strong indicator of glycemic control. Future consideration should be given for tracking metabolic health of young athletes in order to alter training accordingly.

## Figures and Tables

**Figure 1 sports-08-00121-f001:**
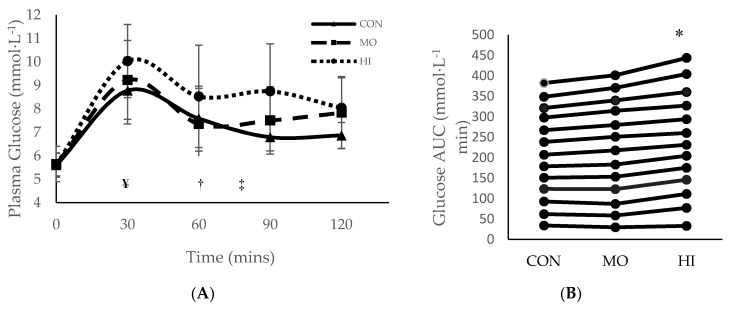
Mean ± SD plasma glucose response for each treatment condition following consumption of glucose solution: (**A**) across all time points; (**B**) two-hour area under the curve (AUC). * Significantly (*p* > 0.05) greater than CON. ¥ Onset of RE in both MO and HI exercise interventions (30 mins); † Completion of MO exercise protocol (60 min); ‡ Completion of HI exercise protocol (75 mins). SD = standard deviation. CON: Control; MO: Moderate Intensity; HI: High Intensity.

**Figure 2 sports-08-00121-f002:**
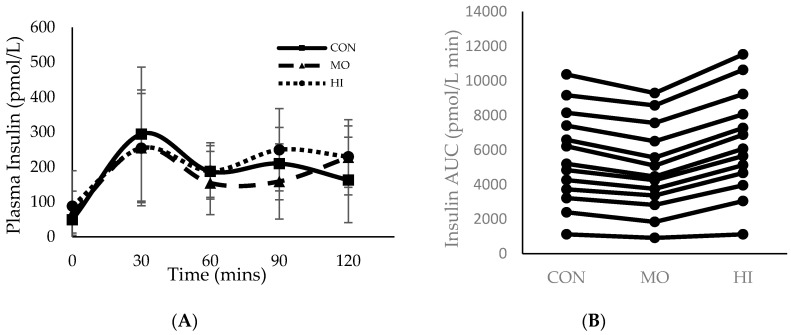
Mean ± SD plasma insulin response for each treatment condition following consumption of glucose solution: (**A**) across all time points; (**B**) two-hour area under the curve (AUC). Onset of RE in both the MO and HI conditions (30 min); Completion of the MO exercise protocol (60 min); Completion of the HI exercise protocol (75 min). CON: Control; MO: Moderate Intensity; HI: High Intensity.

**Figure 3 sports-08-00121-f003:**
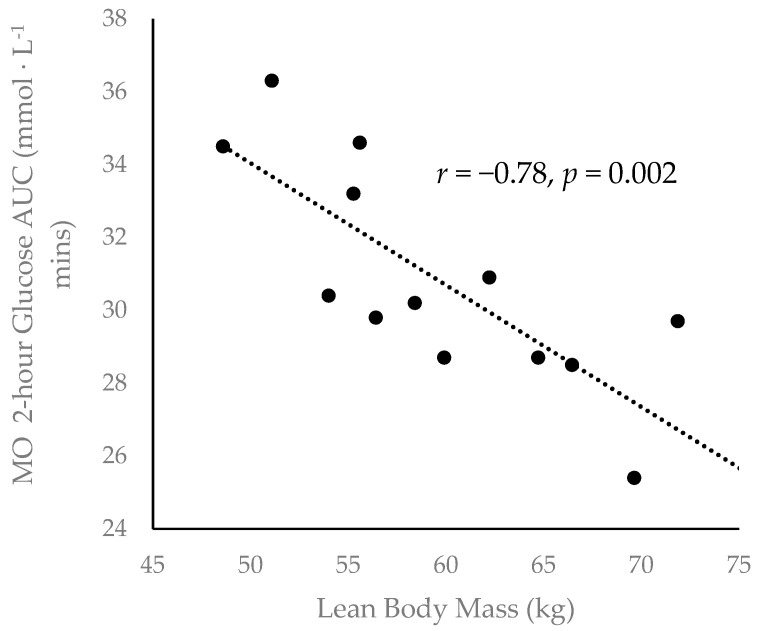
Inverse correlation between participant lean body mass. and two-hour glucose area under the curve (AUC) during the MO condition (*r* = −0.78, *p* = 0.002).

**Table 1 sports-08-00121-t001:** Participant Characteristics (*n* = 13).

Descriptive	Mean ± Std. Deviation
Age (yrs.)	23.43 ± 2.18
Height (cm)	175.16 ± 10.44
Weight (kg)	77.02 ± 8.91
1-RM BHSQ (kg)	116.23 ± 26.25
1-RM BP (kg)	99.17 ± 17.63
Lean Mass Total (kg)	59.55 ± 7.09
Body Fat (%)	20 ± 0.03
HbA1c	5.15 ± 0.17
Fasting Insulin (pmol∙L^−1^)	68.51 ± 19.41

BHSQ: Back-loaded half squat; BP: Bench press; HbA1c: Glycated hemoglobin.

**Table 2 sports-08-00121-t002:** Conditions explained.

Condition	Oral Glucose Dosage	RE Performed 30-min Following Oral Glucose Ingestion
CON	2 g/kg BW	None
MO	2 g/kg BW	3 sets, 14 reps @ 65% 1-RM
HI	2 g/kg BW	5 sets, 4 reps @ 90% 1-RM

CON: Control; MO: Moderate Intensity; HI: High Intensity; BW: bodyweight; g: grams; kg: kilograms; RM: repetition maximum.

**Table 3 sports-08-00121-t003:** Mean plasma glucose levels at fasting and throughout the CON, MO and HI conditions.

Plasma Glucose	Mean ± Std Deviation (mmol∙L^−1^)
Fasting	5.6 ± 0.42
CON 2-hr	7.1 ± 1.3 *
MO 2-hr	7.5 ± 0.6 *
HI 2-hr	8.2 ± 1.9 *^,†^

* significantly greater than fasting glucose (*p* < 0.05); ^†^ significantly greater than CON (*p* < 0.05); CON: Control; MO: Moderate Intensity; HI: High Intensity.

**Table 4 sports-08-00121-t004:** Volume Load (sets × repetitions × load (kg)) across conditions.

Condition	Volume Load Mean ± Std. Deviation
HI BHSQ	2111.89 ± 452.12
HI BP	1776.22 ± 264.38
MO BHSQ	2251.75 ± 462.40
MO BP	1894.41 ± 470.98

CON: Control; MO: Moderate Intensity; HI: High Intensity; BHSQ: Back-loaded half squat; BP: Bench press.

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
