# Peer review of "The Effect of Resistance Exercise Intensity on Acute Hyperglycemia in Young Adult Males"

_sports, 2020, doi:10.3390/sports8090121_

Round 1

Reviewer 1 Report

The purpose of this study was to examine how resistance exercise intensity influences induced hyperglycemia prior to. Authors revealed that high intensity back squat and bench press exercises sustained elevated blood glucose, while it wasn't a case for moderate-intensity resistance exercise as well as in the control group.

General Comment

It seems to me that the authors would like the study to be translated into the real world, although it is not. The authors seem to focus too much on the problem of type 2 diabetes, while the purpose of the study is to assess how the intensity of resistance exercises affect induced hyperglycemia among healthy males. Sure, it should be mentioned but, in my opinion, it’s not the main problem. In the introduction, the authors should review previous studies on the effect of resistance exercise intensity on glucose metabolism. What is the rationale for inducing hyperglycemia in healthy men before training? Maybe, the authors should mention the glycemic index and load? Another thing is, in my opinion, too high-intensity or volume at this intensity, I do not believe that everybody did 5 sets of 4 repetitions at 90%1RM without reaching muscular failure, especially for the participants with 6 months resistance training experience. Further, there is a lack of information about rest intervals between sets. Moreover, why such an intensity? Why the authors choose 90% and 65%1RM? Why such exercises and such volume? Why that amount of glucose pre-workout? Please, provide a rationale for these selections. I’m aware that nowadays, amateurs' awareness of nutrition and supplementation is far from ideal, but who uses 160g of glucose before training? Besides, doing 2 exercises is not typical training. Additionally, there is a lack of hypothesis at the end of the introduction. Briefly, the introduction needs a major revision.

Abstract

Line 8: In my opinion, there should be more details about the participants: weight, BF, BMI, healthy, active men etc., and also under the title "Young, Healthy Adult Men".

Line 12: expand "1-RM"

Line 13: In my opinion, 65% of 1RM should be considered as moderate-intensity

Line 14: How much water was it dissolved in?

Line 24: Keywords should not repeat words in the title

Introduction

Line 27: the “T2DM” abbreviation should be expanded

Line 38-48: what is the point of this paragraph? First, the authors point out that resistance training by increased skeletal muscle mass is beneficial for T2D patients. Then, that RT induces hyperglycemia which may accelerate losses in muscle mass. This makes no sense for me.

Line 49-58: The arguments provided by the authors are, in my opinion, insufficient. How could hyperglycemia affect healthy males or training outcomes? Moreover, the mechanisms for post exercises hyperglycemia effect are largely established, authors should provide this information.

Many details are missing that prevent this protocol from being duplicated by future research. What was the rest interval time between sets? Why bench press and back squat? Lack of details about training routine and warm-up. According to National and Strength and Conditioning Association 1RM table, 4 reps at 90%1RM its max effort, while 14 reps at 65%1RM it's like one rep in reserve, what was the rest intervals? I do not believe that participants did that protocol without reaching muscular failure.

Line 73: there is a lot of signs “@” instead of proper one in the whole manuscript

Line 91: double “each”

Line 92: 1-RM test?

Line 96: and insulin?

Line 106: The authors should add this ref.: 10.2478/hukin-2020-0001 https://www.jssm.org/hf.php?id=jssm-19-317.xml

Discussion

As in the introduction, in my opinion, there should be more information on how resistance training variables can affect glucose metabolism, as well as how the amount of glucose consumed, the glycemic index, the glycemic load, and the tonicity of drinks affecting metabolism, insulin sensitivity, etc. Authors write a lot about diabetics but do not research them. Do the authors believe that a high amount of glucose ingestion before high-intensity training could trigger type 2 diabetes? Moreover, there is a lack of information on study limitations and ideas for future studies

Line 267: I see no point in comparing 160g glucose ingested pre-workout with a commonly drunk energy drink or beverage which contain 10g glucose per 100ml

Line 287: this don’t need an abbreviation

Reviewer 2 Report

Authors should indicate the approval number and the institution to which the Ethics Committee that approved this clinical trial belongs.

Some of the conclusions are speculative and are not based on the results obtained in the work, therefore, they must be modified and exclusively point to the contributions derived from this work, and how they improve current knowledge of the blood glucose level and its relationship with resistance exercise.

Reviewer 3 Report

1. Minor spelling check and journal form.

2. How about using the abbreviation after the first explanation of the abbreviation for T2DM?

Thank you very much.

Reviewer 4 Report

General comments:

I would like to thank the authors for the great effort performed in order to try to improve sport science knowledge in this field.

The present study adds a new insight related to effects of resistance training on hyperglycemia.

The introduction prepares the audience for the research focus and brings some evidences from the literature even if some of the studies are quite old (i.e. 13).

Unfortunately, the experimental procedures are not described at all in the manuscript. Some details just briefly appear in the abstract. It would be impossible to replicate this study as it is written here.

Various words are repeated (i.e. L91), omitted, and/or missing around the document, please check.

Some additional suggestions and comments will be detailed bellow.

Specifics comments:

L27 - T2DM should be defined here and not in L45.

L34 - Some studies should be cited here.

L44 - Please remove the first comma: "In healthy, euglycemic individuals, ...".

L82 - Number of the ethics is missing.

L93 - Please define the standards of the NSCA.

L 95 - CON, LO and HI are not defined.

L97 - BW is not defined before.

L109-110 - As it is described, it is a half-squat not a squat, please change it in the entire manuscript.

L131-133 - This doesn't have to be here as this is not statistics. Please move to experimental procedures once written.

L136-137 - Already written in L97.

L145 - Table 1 should be moved to participants' section. Moreover it should be divided in two tables where statistics between CON, HI and LO would be better explained.

L177 - Table 2 is not clear either. Number of reps, sets and % of 1RM are not presented. Are the results significantly different.

L188 - Fig.1 and Fig.2 are called but Fig.2 doesn't exist in the document. Please check. 

L211 - Please chose Fig.1 or Figure 1 but don't use both. Apply to the entire manuscript.

L232 - both panel of Fig.3 are named (b), please check. Also it should be written A and B not (a) and (b).

L233 - Maybe Fig.2 and not Fig.3?

L251 - Fig.4? If not where is Fig.4?

L252 - Please add "Area under the curve"

L255 - please add "to the best of our knowledge"

L256 - BW or bodyweight? Please chose.

L265-266 Please provide some studies on the topic.

L303 - It is written: "We also contribute data...". Please check words are missing.

L305-307 - What is the point with the data of the present study? Unclear. Please reformulate.

L308-309 - Very speculative. Please provide major studies to illustrate or remove.

Round 2

Reviewer 1 Report

Major:

“This study contains several limitations that should be considered in future research. Firstly, not every subject was able to complete all 14 repetitions in the third set of bench press and BSQ at the moderate intensity level, leading to some variation among subjects in total volume load. We also only used two resistance exercises, and although they both engage large muscle mass, this does not exactly relate to real-world lifting practices. The dosage of glucose used (2g/kg BW) is greater than the sugar content of pre-workout beverages, thus the glycemic response in this study would be expected to be exaggerated compared to a typical pre-workout response. Finally, post-exercise glucose tolerance and insulin sensitivity were not measured, thus the present study only shows a pre, intra and immediately post-workout glycemic response.”

why those things were not considered when designing the study?

Minor:

Line 111 and 595: use abbreviation for resistance exercise

Line 187-188: be consistent which units you use in the whole manuscript, imperial or metric

Reviewer 4 Report

Thanks for taking previous suggestions and recommendations into consideration.

Manuscript looks better now.

Nevertheless, some minor modifications still need to done.

L170 - Please explain BSQ and HbA1c below table 1.

L173 - If you use BSQ why don't you use BP for bench press here? Please change in the entire manuscript.

L174 - BSQ should be renamed BHSQ (standing for Back-loaded HALF squat) if not, it could lead to some misunderstandings

L175 - CON, HI and MO should be defined here and not line 192.

L 259 - Please stipulate the cadence of the repetitions in bench press.

L 262 - Please stipulate the cadence of the repetitions in BHSQ.

L265 - An overall figure to explain the different conditions and times of the present protocol would be vey helpful to help the reader to better understand what has been done in the study. It's a personal point of view.

L295 - Please define CON, MO and HI as it is a title.

L296 - "*significantly greater than fasting glucose (p<.05); †significantly greater than CON (p<.05)." should appear below the table.

L 341 - Please change "Table 3. Volume Load (sets*repetitions*load [Kg]) across conditions."

It should be written : Table 3. Volume Load (sets x repetitions x load [Kg]) across conditions.

L 439 - RE, MO and HI should be defined as it is a figure caption.

L 463-464 - RE, MO and HI should be defined as it is a figure caption.

L 522 - Please define MO

L 522-523 - r = -0.78, p = 0.002 should appear in the graph. Please change. 

By the way, why don't you show HI conditions to illustrate differences between HI and MO?

Again just my own point of view to allow the reader to better understand your work.

L 534 - "the insults of diabetes" --> insulin maybe?

L 549 - "as others have" please indicate some studies.

L 552-553 Why HPG stands for Hepatic Glucose Production? Shouldn't it be HGP? It is confusing. Please change it in the manuscript.

Thanks for your work.
